# Incremental Detection Rate of Dysplasia and Sessile Serrated Polyps/Adenomas Using Narrow-Band Imaging and Dye Spray Chromoendoscopy in Addition to High-Definition Endoscopy in Patients with Long-Standing Extensive Ulcerative Colitis: Segmental Tandem Endoscopic Study

**DOI:** 10.3390/diagnostics13030516

**Published:** 2023-01-31

**Authors:** Ji Eun Kim, Chang Wan Choi, Sung Noh Hong, Joo Hye Song, Eun Ran Kim, Dong Kyung Chang, Young-Ho Kim

**Affiliations:** Department of Medicine, Samsung Medical Center, Sungkyunkwan University School of Medicine, Seoul 60351, Republic of Korea

**Keywords:** ulcerative colitis, sessile serrated lesion, high-definition endoscopy, chromoendoscopy, narrow band imaging

## Abstract

High-definition (HD) endoscopy is recommended in surveillance colonoscopy for detecting dysplasia in patients with ulcerative colitis (UC). Dye-spray chromoendoscopy (DCE) and narrow-band imaging (NBI) are often used as adjunctive techniques of white-light endoscopy (WLE) in real-world practice. However, the incremental detection ability of DCE and NBI added to HD-WLE for dysplasia and serrated lesions has not yet been evaluated using tandem endoscopy in patients with long-standing extensive UC. We enrolled patients with extensive UC for >8 years who were in clinical remission (partial Mayo score < 2) at the Samsung Medical Center in Seoul, Republic of Korea. HD-WLE was performed first. Subsequently, HD-NBI and HD-DCE with indigo carmine were performed using the segmental tandem colonoscopy technique. A total of 40 patients were eligible, and data obtained from 33 patients were analyzed. The incremental detection rates (IDRs) for dysplasia and serrated lesions were calculated. HD-WLE detected three dysplasia and five sessile serrated adenomas/polyps (SSAs/Ps). HD-NBI and HD-DCE did not detect additional dysplasia (IDR = 0%; 95% confidence interval (CI): 0–56.2%). HD-NBI identified one missed SSA/P (IDR = 7.7%; 95% CI: 1.4–33.3%), and HD-DCE detected seven missed SSAs/Ps (IDR = 53.9%; 95% CI: 29.1–76.8%). Logistic regression found that HD-DCE increased the detection of SSAs/Ps compared to HD-WLE and/or HD-NBI (odds ratio (OR) = 3.16, 95% CI: 0.83–11.92, *p* = 0.08). DCE in addition to HD-WLE improved the detection of SSAs/Ps, but not dysplasia, in patients with long-standing extensive UC.

## 1. Introduction

Colonoscopic surveillance in patients with inflammatory bowel disease (IBD) is associated with a reduced risk of colorectal cancer (CRC) and CRC-associated mortality [1]. Professional societies advocate for surveillance colonoscopy with high-definition (HD) endoscopy or standard-definition (SD) endoscopy with dye-spray chromoendoscopy (DCE) to optimize dysplasia detection [2,3,4,5,6,7,8]. The clear trend in diagnosing more colitis-associated dysplasia might be attributed to the awareness of the increased risk of CRC and the dissemination of routine surveillance in patients with long-standing IBD [4,9].

In clinical practice, DCE seems to be poorly accepted due to its longer examination time and technical inconvenience [10,11]. Presently, HD endoscopes are widely distributed in clinical settings. HD white-light endoscopy (WLE) with targeted biopsies is considered the technique of choice in clinical practice and in most recent clinical trials. DCE is strongly recommended when performing surveillance with SD colonoscopies [4]. However, it is unclear whether DCE and narrow-band imaging (NBI) are still beneficial when surveillance is performed along with HD colonoscopy. A recent meta-analysis demonstrated that the diagnostic yield of DCE for dysplasia was superior to that of HD-WLE in non-randomized controlled trials, despite failed randomized controlled trials (RCTs) [12].

The role of the serrated pathway in the tumorigenesis of IBD-related CRC remains unclear [13]. The serrated pathway is thought to be associated with up to 30% of sporadic CRC. Serrated lesions are often subtle, pale in appearance, and easy to miss. DCE significantly improved the detection of proximal serrated lesions compared to standard WLE [14]. Serrated lesions are not uncommon in the IBD population [13]. Therefore, contrast-enhanced endoscopic techniques may improve the ability to detect any dysplastic lesions, as well as serrated lesions, during surveillance colonoscopy.

In real-world practice, unlike clinical trials, DCE and NBI are performed during surveillance as adjunctive techniques for WLE. Thus, the objective of this study was to evaluate the incremental detection rate for dysplasia and sessile serrated lesions by adjunctive DCE and NBI with HD-WLE using segmental tandem colonoscopy in patients with long-standing extensive UC.

## 2. Materials and Methods

### 2.1. Study Population

This single-center prospective observational study was conducted on consecutive adult patients with long-standing extensive UC who underwent surveillance colonoscopies from January to December 2021 at the Samsung Medical Center, Seoul, Republic of Korea. The eligibility criteria were: (1) age ≥ 18 years, (2) extensive colitis (affecting beyond the splenic flexure), (3) ≥8 years since a UC diagnosis, and (4) clinical remission (partial Mayo score < 2). The exclusion criteria were as follows: (1) prior colonoscopy < 12 months earlier, (2) partial or total colectomy, (3) inadequate bowel preparation (Boston Bowel Preparation Scale score < 6), (4) failure of cecal intubation, (5) incomplete performance of NBI and DCE, (6) coagulopathy (patients taking dual anti-platelet agents and warfarin, platelet count < 100,000/mm^3^, and/or prothrombin time international normalized ratio > 1.4), and (7) allergy to indigo carmine.

### 2.2. Segmental Tandem Colonoscopy

All colonoscopies were performed by a single experienced endoscopist (S.N.H., who has performed more than 5000 colonoscopies and more than 500 colonoscopies in patients with UC) using HD endoscopes (CF-H260AI/L, Olympus, Tokyo, Japan). The colonoscope was advanced to the cecal end. Long-standing extensive UC induces colon length shortening, disturbing the clear discrimination of colonic segments. Therefore, we divided the colon into three equal lengths (proximal, mid, and distal colon). Each segment of the colon was observed in three sequential phases with the insertion and withdrawal of the colonoscope to detect mucosal lesions. In the first phase, HD-WLE was performed to detect dysplasia and serrated lesions. In the second phase, virtual chromoendoscopy (VCE) was performed using NBI. In the third phase, DCE was performed with indigo carmine spray. An HD colonoscope was inserted under white light, and the lumen in each part of the colon was sprayed with 0.4% indigo carmine delivered via a dye spray catheter (PW-5V1, Olympus). After allowing a few seconds for the dye to settle onto the mucosal surface, excess pools of indigo carmine were suctioned. The HD colonoscope was re-inserted, and the mucosa was then examined. The withdrawal time from each part of the colon was measured using a stopwatch and excluded the time used for insertion, washing, biopsy, and polypectomy.

Detected lesions were biopsied or removed after finishing the three phases of the examination. Lesions < 5 mm in the rectosigmoid colon with a high degree of confidence as hyperplastic polyps and those clearly suggestive of pseudo-polyps were not removed. Lesions < 5 mm were removed using biopsy forceps. Lesions ≥ 5 mm were resected using a polypectomy snare with cold snare polypectomy or the endoscopic mucosal resection technique.

The quality of bowel cleansing was graded using the Boston Bowel Preparation Scale [15]. Endoscopic disease activity was evaluated by Mayo endoscopic sub-scores for UC [16]. The extent of involvement of UC was estimated based on colonoscopic features, comprising scarring, tubular colonic appearance, featureless colon, and the presence of post-inflammatory polyps and colonic strictures [17].

### 2.3. Pathologic Interpretation

The pathology of the targeted biopsy or resected specimens was reviewed by board-certified pathologists. Pathologic diagnoses for colitis-associated dysplasia were categorized as negative for dysplasia, indefinite for dysplasia (IND), low-grade dysplasia (LGD), high-grade dysplasia (HGD), and adenocarcinoma [18,19]. No lesion was classified as IND. Any discrete adenomatous lesion located at a non-colitic segment was categorized as a sporadic adenoma. Serrated lesions were classified as hyperplastic polyps (HPs), sessile serrated adenomas/polyps (SSAs/Ps), and traditional serrated adenomas (TSAs) [19]. No lesion was diagnosed as TSA.

### 2.4. Outcome Measures

For analysis purposes, only those lesions located within disease areas were included and then classified into three groups: (1) dysplasia, which included LGD, HGD, and adenocarcinomas, (2) SSAs/Ps, and (3) non-dysplastic lesions, which included HPs < 10 mm, pseudo-polyps, scar tissue, and other unspecific and non-neoplastic mucosal changes.

In the per-patient analysis, the detection rate of dysplasia, serrated lesions, and any lesions suspected of dysplasia was defined as the proportion of patients who had at least one dysplastic, SSA/P, or non-dysplastic lesion in relation to the total number of screened patients. In the per-lesion analysis, the detection rate of dysplasia, SSA/P, and non-dysplastic lesions was defined as the proportion of dysplasia, SSA/P, and non-dysplastic lesions in relation to the total number of dysplasia, SSA/P, and non-dysplastic lesions. To evaluate the additional effect of HD-NBI and HD-DCE, the incremental detection rate of dysplasia, serrated lesions, and any lesions was calculated.

### 2.5. Sample Size Calculation

Based on a previous study, the dysplasia detection rate of WLE was estimated to be 6.5% [20,21,22,23,24,25,26], and that of HD-DCE was estimated to be 20.6% [26]. In this study design, a single study cohort was compared to a known value, and the primary endpoint was with a binary primary endpoint (presence versus absence of lesion). Type I error and power were set at 0.05 and 0.8, respectively. The calculated sample size was 34. Considering a dropout rate of 20%, we decided to enroll 40 patients (https://clincalc.com/stats/samplesize.aspx, accessed on 1 October 2020).

### 2.6. Statistical Analysis

All statistical analyses were performed with GraphPad Prism 9.3.1 (GraphPad Software, Inc., San Diego, CA, USA). Quantitative variables are expressed as means ± standard deviation (SD). Categorical variables are expressed as total numbers and frequencies (%). The quantitative variables were analyzed using the Student’s *t*-test or the Mann–Whitney test, as appropriate. Qualitative variables were analyzed using the χ^2^ test or Fisher’s exact test as appropriate. Logistic regression analysis with Firth’s penalized likelihood approach was performed to identify the factors associated with lesions missed by HD-WLE.

## 3. Results

### 3.1. Patient Characteristics

A total of 40 patients were assessed for eligibility. Seven patients were excluded due to residual active inflammation (*n* = 5) and inadequate bowel preparation (*n* = 2). A total of 33 patients were included in the study (mean age = 58.7 ± 13.7 years; male, *n* = 22). The mean duration of UC before surveillance colonoscopy was 14.6 ± 5.9 years. The disease extended into the cecum (*n* = 5), ascending colon (*n* = 11), hepatic flexure (*n* = 3), and transverse colon (*n* = 14) (Table 1).

The insertion time to the cecal end was 3.2 ± 2.1 min. The withdrawal time of HD-WLE, HD-NBI, and HD-DCE from the entire colon was 6.5 ± 1.1, 6.2 ± 2.0, and 9.2 ± 3.4 min, respectively. Figure 1 illustrates the study flow diagram.

### 3.2. Per-Polyp Analysis of Dysplasia, SSAs/Ps, and Non-Dysplastic Lesions Detected by HD-WLE, HD-NBI, and HD-DCE

Three dysplastic lesions were detected in UC-involved colonic areas, all of which were detected by HD-WLE (Figure 2A). Among the three dysplastic lesions, one was located in the ascending colon, and two were located in the transverse colon.

The total number of detected SSAs/Ps was 13, of which two SSA/Ps (15.4%) were located in the proximal colon, three (23.1%) were in the mid-colon, and eight (61.5%) were in the distal colon (Figure 2B). Among these 13 SSAs/Ps, five lesions (38.5%) were detected by HD-WLE (Figure 3A), one lesion was detected by HD-NBI (Figure 3B), and seven SSAs/Ps were detected by HD-DCE (Figure 3C). The incremental detection rates for dysplasia and SSAs/Ps by NBI were 0% (95% CI: 0–56.2%) and 7.7% (95% CI: 1.4–33.3%), respectively. The incremental detection rates for dysplasia and SSAs/Ps by DCE were 0% (95% CI: 0–56.2%) and 53.9% (95% CI: 29.1–76.8%), respectively.

The total number of non-dysplastic lesions was 40, of which 21 (52.5%), 6 (15.0%), and 13 (32.5%) were detected by HD-WLE, HD-NBI, and HD-DCE, respectively (Table 2). About half of the non-dysplastic lesions were detected by HD-NBI and HD-DCE. The sporadic lesions that developed in the colonic area without UC involvement included one adenoma, one SSA/P, and eight non-dysplastic lesions. In this study, sporadic lesions were excluded from the analysis due to their lack of association with colitis.

In logistic regression analysis, HD-WLE was associated with an increased risk of missed SSAs/Ps (OR = 3.16, 95% CI: 0.83–11.92, *p* = 0.08) (Table 3).

### 3.3. Per-Patient Analysis of Dysplasia, SSAs/Ps, and Non-Dysplastic Lesions Detected by HD-WLE, HD-NBI, and HD-DCE

The numbers of patients with one or more colitis-associated dysplastic, SSA/P, or non-dysplastic lesions was 3 (9.1%), 11 (33.3%), and 19 (57.6%), respectively. HD-NBI and HD-DCE could not identify additional patients with dysplasia. In contrast, HD-NBI identified one additional patient with an SSA/P, and HD-DCE identified seven additional patients with SSAs/Ps. The incremental detection rate of SSAs/Ps by HD-NBI and HD-DCE was 9% (95% CI: 16.2–37.7%) and 63.6% (95% CI: 35.4–84.8%), respectively. One patient had both dysplasia in the ascending colon and two SSAs/Ps in the transverse colon and rectum. The number of patients with one or more colitis-associated neoplasia, including dysplasia and SSAs/Ps, was 10 (30.3%). HD-NBI identified one additional patient, and HD-DCE identified three patients whose SSAs/Ps were missed by HD-WLE (Table 4).

In univariate logistic regression, patients with multiple SSAs/Ps who underwent only HD-WLE showed an increased risk for missed dysplasia and/or SSAs/Ps. However, in multivariate logistic regression adjusted for univariate-identified clinical factors (*p* < 0.2), no clinical factors could predict the patients with missed dysplasia and/or SSAs/Ps (Table 5).

## 4. Discussion

The three-year post-colonoscopy CRC rate in patients with IBD was reported to be much higher than that in the general population [27]. This means that IBD-associated CRC developed through a dysplasia-carcinoma sequence, where there may have been an opportunity to diagnose CRC at an earlier stage or to prevent it altogether if more precise and sensitive techniques to detect dysplasia, such as DCE and NBI, were used [28]. The benefit of DCE and VCE in addition to SD colonoscopy has been clearly demonstrated. However, the benefit of DCE and VCE in addition to HD endoscopy is unclear. We evaluated the incremental detection rate of HD-NBI and HD-DCE over HD-WLE using segmental tandem colonoscopy. DCE as an adjunct to HD-WLE improved the detection of SSAs/Ps in patients with long-standing extensive UC.

A Spanish multicenter study reported that the incremental detection yield for dysplasia by DCE was 57.4% (95% CI: 47.5% to 67.3%) [29]. However, incremental diagnostic yields for SSAs/Ps by NBI and DCE have not been reported yet. In this study, we found that the incremental detection rates for SSAs/Ps by NBI and DCE were 8% and 54% in per-polyp analysis and 9% and 64% in per-person analysis, respectively. Limited studies have examined serrated lesions within the IBD population. A retrospective cohort study reported that 134 IBD patients with SSAs/Ps had a heightened risk of synchronous multifocal visible dysplasia (16% vs. 3%), without showing a greater risk of metachronous dysplasia compared to non-IBD patients with SSAs/Ps [30]. Another retrospective study of 78 serrated polyps reported that the 10-year rates of incidence of advanced neoplasia in patients with low-grade dysplasia, indefinite for dysplasia, and negative for dysplasia were 17, 8, and 0%, respectively [31]. A recently published retrospective study of 621 IBD patients reported no difference in CRC risk between patients with hyperplastic polyps and sessile serrated lesions without dysplasia. However, TSAs and sessile serrated lesions with dysplasia were associated with subsequent advanced colorectal neoplasia [32]. However, previous studies adopted different patient selection criteria, methodologies, histopathologic criteria, and outcome measures. In addition, there was no difference in the rate of synchronous or metachronous dysplasia in serrated lesions found in inflamed versus noninflamed mucosa, suggesting that it is unclear whether inflammation is associated with serrated lesions [13]. In this study, about one-fourth of the patients with long-standing UC had SSAs/Ps. Considering that SSAs/Ps and TSAs might have malignant potential and increase the risk of subsequent CRC, efforts to detect serrated lesions in patients with IBD should be required. DCE may be one of the best options to improve the detection of SSAs/Ps in long-standing extensive UC patients.

In this study, HD-NBI and HD-DCE failed to increase the detection rate of dysplasia compared to HD-WLE. The diagnostic yield of DCE was mainly evaluated in studies using SD endoscopy. The value of DCE as an adjunct to HD endoscopy has not been well studied. A recent Korean multicenter RCT evaluated the detection rates of colitis-associated dysplasia between HD-WLE with random biopsies and HD-DCE with targeted biopsies [26] and found that HD-DCE showed improved dysplasia detection compared to HD-WLE (20.6% vs. 12.0%). However, the difference failed to reach statistical significance. In RCTs using random biopsies, HD-DCE was superior to HD-WLE for detecting dysplasia (11.1% vs. 4.5%) [33]. Though HD-DCE may provide higher detection rates for dysplasia compared to HD-WLE [26,33], there are several barriers to performing DCE in routine practice. First, although DCE is a simple technique and is easy to learn, expertise might affect the outcomes [34]. Second, the equipment required to perform DCE includes indigo carmine or methylene blue. Spray catheters or an infusion pump is needed to spray a uniform mist of the staining agent. Third, additional procedure-related time is needed to perform high-quality DCE for quantification. In a meta-analysis, DCE increased the procedure time by 11 min overall [35]. In this study, DCE needed an additional 9.5 min, excluding the time used for dye spraying, excessive dye suction, and biopsies. There is an ongoing debate about the best technique for surveillance colonoscopy to detect dysplasia in patients with long-standing UC.

VCE is an image-enhancing endoscopic technology used in conjunction with high-definition imaging provided by endoscope manufacturers, including NBI, iScan, and FICE [36]. Recent VCE technology has shown improvements by applying filters with brighter contrast compared to first-generation VCE. Early studies applying first-generation VCE technology showed no additional benefit for the diagnostic yield of colitis-associated dysplasia compared to WLE. However, RCTs applying recent VCE technology showed that VCE was non-inferior to HD-WLE and DCE [37,38]. VCE has several advantages over DCE, such as a shorter procedure time and lack of required sprays or dyes [39]; however, despite technical convenience, NBI did not enhance the detection of dysplasia or serrated lesions in the present study or previous studies [40,41,42].

Previous studies have demonstrated that DCE improved colorectal neoplasia detection in high-risk patients with long-standing IBD, and VCE was shown to improve the characterization of diminutive colorectal lesions [43,44,45,46,47,48]. A recent network meta-analysis identified the significant superiority of DCE to WLE in detecting dysplasia (OR = 2.12), which was similar to our findings [45]. Because of their narrow scanning area, DCE and VCE are best used in conjunction with WLE in routine practice. The best combination for IBD surveillance appears to be WLE for the identification of suspicious areas, with a further examination by DCE to detect superficial colorectal neoplasia [46]. However, cost, availability, time, and experience are still issues. Therefore, evolving technology in combination with HD-WLE can be applicable in selected cases in routine clinical practice, enabling further definition of the lesions and the assessment of their histology, and thus facilitating real-time in vivo diagnoses and decisions regarding the resection of lesions [49]. Furthermore, VCE techniques were superior to WLE for assessing the activity and extent of colorectal IBD, which can bring additional benefits in assessing mucosal healing [44].

Recently, novel endoscopy techniques have emerged [48]. Confocal laser endomicroscopy (CLE) can be used to characterize a lesion, providing the same results as conventional histology [47], so in vivo microscopic assessment of the colonic mucosa may be possible [49]. Devices to improve endoscopic stabilization and visualization, such as Endocuff, may facilitate the detection and removal of colorectal neoplasia, especially in the flexible folds of the sigmoid colon [50]. Recently, non-blinded and randomized controlled trials showed that computer-aided detection improved the detection of colorectal neoplasia by providing visual alarms during the procedure [51,52]. Most studies evaluating the efficacy of newly developed devices in improving the detection of colorectal neoplasia tended to exclude patients with IBD. Therefore, the question of whether new technology such as the Endocuff and computer-aided detection improves UC surveillance should be studied.

This study had several limitations. First, the sample size was relatively small, even though a sample size calculation was performed. Second, this study was conducted at a single academic institution by a single endoscopist. Third, the study design was not a randomized controlled trial, although the tandem endoscopic design is considered a reliable method for evaluating different diagnostic modalities [36]. Therefore, one should be careful when interpreting or extrapolating these data. Further prospective studies with a larger population are needed. Fourth, when the colon was divided into three parts, the cecum and ascending colon were considered the proximal colon, the transverse and descending colon were considered the mid-colon, and the sigmoid colon and rectum were considered the distal colon in healthy controls. However, long-standing extensive UC induces shortening of the colon length, diffuse cicatricial change, and lead pipe appearance, disturbing clear discrimination of the colonic segments, especially the sigmoid-descending (SD) junction, the boundary of the mid and distal colon. Therefore, this study divided the colon into three equal lengths. In addition, we measured the withdrawal time excluding the time used for washing, biopsy, and polypectomy, which indicated the pure observation time. The withdrawal time of HD-WLE and HD-NBI during colonoscopy was just over six minutes. However, in practice, withdrawal time includes the time for washing, biopsy, and polypectomy during colonoscopy withdrawal. The total withdrawal time, including washing, biopsy, and polypectomy, was 33.5 ± 12.1 min. When simply calculated, the total withdrawal time of HD-WLE, HD-NBI, and HD-DCE was estimated to be more than 10 min.

## 5. Conclusions

A meticulous examination is needed to reduce the CRC deaths of IBD patients. Colitis-associated neoplasia is often flat and subtle, making it difficult to differentiate from the surrounding scarring and inflamed mucosa. In the era of HD endoscopy, NBI might have limited value in UC surveillance colonoscopy. However, DCE improved the detection of SSAs/Ps in patients with long-standing extensive UC at rates as much as 54%. A few studies evaluated the risk of CRC associated with serrated lesions in IBD. The histologic classification of serrated lesions is being established, and larger studies are required to evaluate the extent to which serrated lesions are involved in the development of colitic cancer in IBD patients. Therefore, our results indicating that DCE has an advantage in detecting serrated lesions in patients with long-standing UC have clinical implications.

## Figures and Tables

**Figure 1 diagnostics-13-00516-f001:**
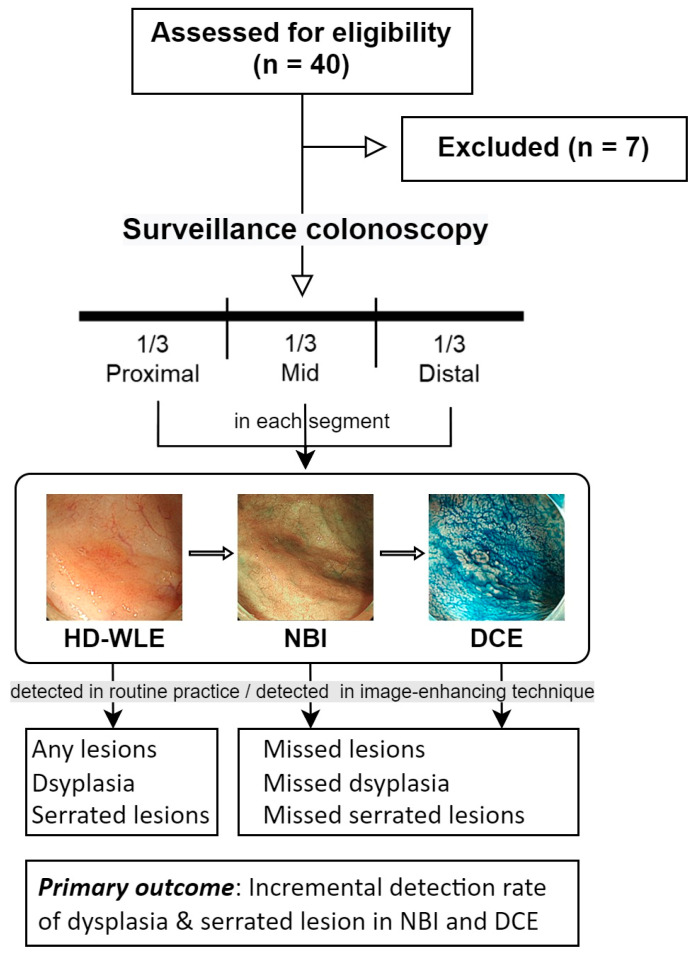
Study flow diagram. HD-WLE, high-definition white-light endoscopy; NBI, narrow-band imaging; DCE, dye spray chromoendoscopy.

**Figure 2 diagnostics-13-00516-f002:**
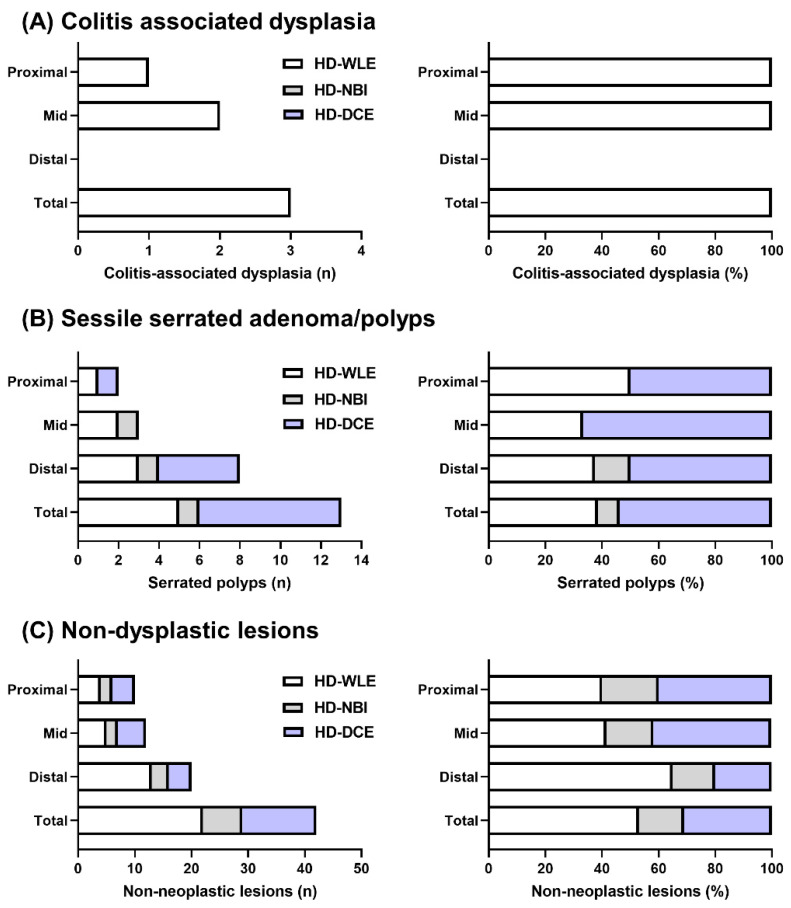
Number and proportion of dysplastic lesions, sessile serrated adenomas/polyps, and nondysplastic lesions detected by high-definition white-light endoscopy (HD-WLE), high-definition narrow-band imaging (HD-NBI), and high-definition dye spray chromoendoscopy (HD-DCE). (**A**) Colitis-associated dysplasia, (**B**) sessile serrated adenomas/polyps, and (**C**) non-dysplastic lesions. HD-WLE, high-definition white-light endoscopy; HD-NBI, high-definition narrow-band imaging; HD-DCE, high-definition dye spray chromoendoscopy.

**Figure 3 diagnostics-13-00516-f003:**
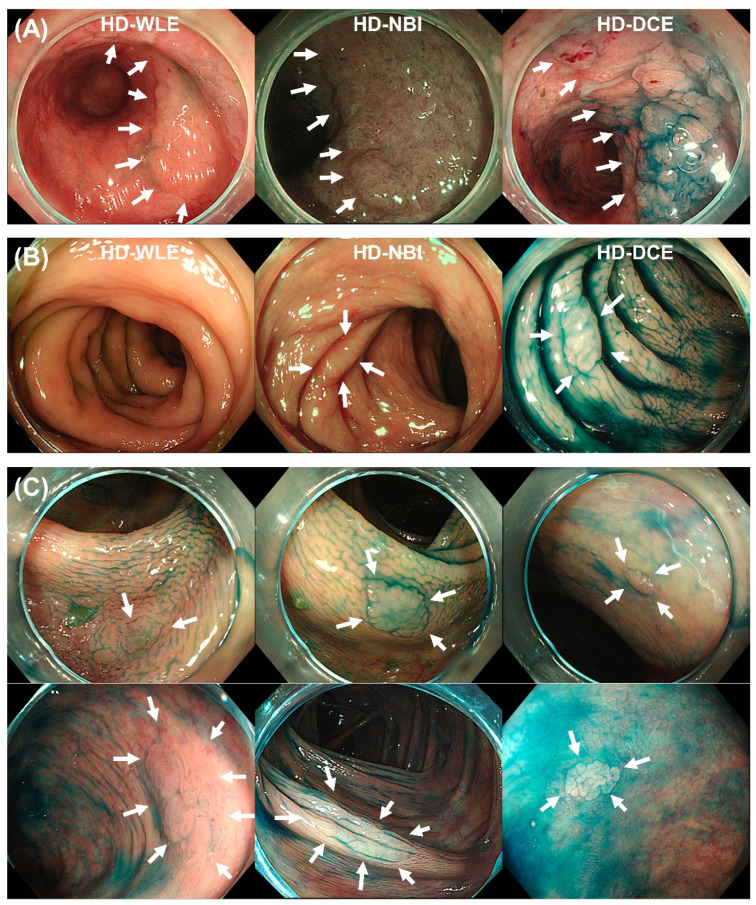
Sessile serrated adenomas/polyps detected by high-definition white-light endoscopy (HD-WLE), high-definition narrow-band imaging (HD-NBI), and high-definition dye spray chromoendoscopy (HD-DCE). (**A**) Sessile serrated adenoma/polyp detected in HD-WLE, (**B**) sessile serrated adenoma/polyp detected in HD-NBI, and (**C**) sessile serrated adenoma/polyp detected in HD-DCE. HD-WLE, high-definition white-light endoscopy; HD-NBI, high-definition narrow-band imaging; HD-DCE, high-definition dye spray chromoendoscopy.

**Table 1 diagnostics-13-00516-t001:** Baseline characteristics of study subjects.

Variables	Value	Variables		Value
Age at UC onset, years (mean ± SD)	44.1 ± 16.3	UC extension, *n* (%)		
Age at surveillance, years (mean ± SD)	58.7 ± 13.5	Cecum		5 (15.2)
Male, *n* (%)	23 (67)	Transverse colon		11 (33.3)
Disease duration, years (mean ± SD)	14.6 ± 5.9	Hepatic flexure		3 (9.1)
Partial Mayo score, *n* (%)		Ascending colon		14 (42.4)
0	19 (57.6)	Mayo endoscopic sub-score, *n* (%)		
1	14 (42.4)	0		15 (45.5)
Hemoglobin, g/dL (mean ± SD)	14.1 ± 1.3	1		18 (54.5)
Albumin, g/dL (mean ± SD)	4.4 ± 0.3	UCEIS, point (mean ± SD)		2.1 ± 2.1
C-reactive protein, mg/dL (mean ± SD)	0.23 ± 0.5	Boston bowel preparation scale		8.9 ± 0.3
Medication, *n* (%)				
5-Aminosalicylates	31 (93.9)	Withdrawal time, min (mean ± SD)		
Azathioprine	11 (33.3)	Proximal colon	HD-WLE	2.6 ± 0.8
Infliximab	10 (30.3)		HD-NBI	2.4 ± 2.0
Adalimumab	2 (6.1)		HD-DCE	3.6 ± 2.6
Vedolizumab	3 (9.1)	Mid colon	HD-WLE	1.9 ± 0.6
Primary sclerosing cholangitis, *n* (%)	1 (3.0)		HD-NBI	2.2 ± 0.7
CRC history in 1st degree relative, *n* (%)	0		HD-DCE	2.4 ± 1.1
No. of surveillance colonoscopies, *n* (mean ± SD)	5.7 ± 4.0	Distal colon	HD-WLE	2.0 ± 1.4
Interval from prior colonoscopy, months (mean ± SD)	18.4 ± 10.7		HD-NBI	1.6 ± 2.1
Previous history of dysplasia and CRC, *n* (%)	5 (15.2)		HD-DCE	3.5 ± 2.7

CRC colorectal cancer, UCEIS Ulcerative Colitis Endoscopic Index of Severity, HD-WLE high-definition white-light endoscopy, HD-NBI high-definition narrow-band imaging, HD-DCE high-definition dye spray chromoendoscopy.

**Table 2 diagnostics-13-00516-t002:** Per-polyp analysis of dysplasia, sessile serrated adenoma/polyps, and non-dysplastic lesions detected by HD-WLE, HD-NBI, and HD-DCE.

	Overall						Proximal colon					
	HD-WLE	HD-NBI		HD-DCE		Total	HD-WLE	HD-NBI		HD-DCE		Total
	*n*	*n*	IDR	*n*	IDR	*n*	*n*	*n*	IDR	*n*	IDR	*n*
Dysplasia	3	0	0%	0	0%	3	1	0	0%	0	0%	1
SSAs/Ps	5	1	8%	7	54%	13	1	0	0%	1	50%	2
Non-dysplastic lesions	21	6	15%	13	33%	40	3	1	13%	4	50%	8
	**Mid-colon**						**Distal colon**					
	**HD-WLE**	**HD-NBI**		**HD-DCE**		**Total**	**HD-WLE**	**HD-NBI**		**HD-DCE**		**Total**
	** *n* **	** *n* **	**IDR**	** *n* **	**IDR**	** *n* **	** *n* **	** *n* **	**IDR**	** *n* **	**IDR**	** *n* **
Dysplasia	2	0	0%	0	0%	2	0	0	-	0	-	0
SSAs/Ps	1	0	0%	2	67%	3	3	1	13%	4	50%	8
Non-dysplastic lesions	5	2	17%	5	42%	12	13	3	15%	4	20%	20

IDR, incremental detection rate; HD-WLE, high-definition white-light endoscopy; HD-NBI, high-definition narrow-band imaging; HD-DCE, high-definition dye spray chromoendoscopy; SSA/Ps, sessile serrated adenoma/polyps.

**Table 3 diagnostics-13-00516-t003:** Univariate and multivariate logistic regression for lesions missed by HD-WLE.

	Univariate			Multivariate		
	OR	95% CI	*p*	OR	95% CI	*p*
Dysplasia	0.23	0.01–5.09	0.35	0.17	0.01–3.01	0.22
Sessile serrated adenoma/polyp	2.69	0.75–9.58	0.12	3.16	0.83–11.92	0.08
Non-dysplastic lesion	0.61	0.18–2.03	0.42	0.61	0.18–2.03	0.42

HD-WLE, high-definition white-light endoscopy; HD-NBI, high-definition narrow-band imaging; HD-DCE, high-definition dye spray chromoendoscopy; OR, odds ratio; CI, confidence interval.

**Table 4 diagnostics-13-00516-t004:** Per-patient analysis of dysplasia, sessile serrated adenoma/polyps, and non-dysplastic lesions detected by HD-WLE, HD-NBI, and HD-DCE.

	Overall						Proximal colon					
	HD-WLE	HD-NBI		HD-DCE		Total	HD-WLE	HD-NBI		HD-DCE		Total
	*n*	*n*	IDR	*n*	IDR	*n*	*n*	*n*	IDR	*n*	IDR	*n*
Dysplasia	3	0	0%	0	0%	3	1	0	0%	0	0%	1
SSAs/Ps	4	1	9%	7	64%	11	1	0	0%	1	50%	2
Non-dysplastic lesions	10	5	26%	9	47%	19	3	1	25%	2	50%	4
	**Mid-colon**						**Distal colon**					
	**HD-WLE**	**HD-NBI**		**HD-DCE**		**Total**	**HD-WLE**	**HD-NBI**		**HD-DCE**		**Total**
	** *n* **	** *n* **	**IDR**	** *n* **	**IDR**	** *n* **	** *n* **	** *n* **	**IDR**	** *n* **	**IDR**	** *n* **
Dysplasia	2	0	0%	0	0%	2	0	0	-	0	-	0
SSAs/Ps	1	0	0%	2	67%	3	3	1	14%	4	57%	7
Non-dysplastic lesions	3	1	11%	5	56%	9	6	3	38%	2	25%	8

*IDR,* incremental detection rate; *HD-WLE,* high-definition white-light endoscopy; *HD-NBI,* high-definition narrow-band imaging; *HD-DCE,* high-definition dye spray chromoendoscopy; *SSA/Ps*, sessile serrated adenoma/polyps.

**Table 5 diagnostics-13-00516-t005:** Univariate and multivariate logistic regression analysis for factors associated with missed dysplasia and/or sessile serrated adenoma/polyps (SSAs/Ps).

	Univariate				Multivariate	
	OR	95% CI	*p*-Value	OR	95% CI	*p*-Value
Age (+1 year)	1.01	0.95–1.06	0.81			
Male (vs. Female)	2.23	0.41–12.11	0.35			
No. of prior surveillance conolonoscopies	0.99	0.81–0.19	0.89			
Interval from prior colonoscopy (+1 month)	0.98	0.91–1.06	0.65			
5-Aminosalicylates	2.44	0.06–108.35	0.64			
Azathioprine	0.85	0.18–4.12	0.84			
Infliximab	1.03	0.21–5.07	0.97			
Adalimumab/vedolizumab	0.17	0.02–1.23	0.079	0.12	0.01–2.04	0.14
Withdrawal time (+1 min)	1.08	0.95–1.24	0.24			
Large SSA/P (>1 cm)	7.00	0.70–69.68	0.097	10.28	0.21–496.67	0.24
Multiple SSAs/Ps (≥2)	14.99	1.72–131.21	0.014	4.07	0.35–47.27	0.26
Previous dysplasia	4.01	0.56–28.74	0.17	2.81	0.24–32.80	0.41

## Data Availability

The data presented in this study are available upon request from the corresponding author.

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
