# Peer review of "Incremental Detection Rate of Dysplasia and Sessile Serrated Polyps/Adenomas Using Narrow-Band Imaging and Dye Spray Chromoendoscopy in Addition to High-Definition Endoscopy in Patients with Long-Standing Extensive Ulcerative Colitis: Segmental Tandem Endoscopic Study"

_diagnostics, 2023, doi:10.3390/diagnostics13030516_

Round 1
Reviewer 1 Report
Dear Authors, I think that this study should be improved. In particular, some references should be added to bibliography: “AGA Clinical Practice Update on Endoscopic Surveillance and Management of Colorectal Dysplasia in Inflammatory Bowel Diseases: Expert Review”, “Third European Evidence-based Consensus on Diagnosis and Management of Ulcerative Colitis. Part 1: Definitions, Diagnosis, Extra-intestinal Manifestations, Pregnancy, Cancer Surveillance, Surgery, and Ileo-anal Pouch Disorders”. The sample saze I stoo small to derive reliable results. Other limits of the study are the monocentric nature, the lack of randomization to compare results and use of a single endoscopist.Finally, I don’t understand why the colon segments have been divided into three equal parts in the evaluation of surveillance.
Author Response
Revision Point-by-Point
Thank you very much for your generous and discerning comments and kind suggestions regarding our manuscript. We made our best efforts to revise the manuscript according to your comments and recommendations. The point-by-point responses to your comments and suggestions are shown below.
Answers to comments from the Reviewer #1
- In particular, some references should be added to bibliography: “AGA Clinical Practice Update on Endoscopic Surveillance and Management of Colorectal Dysplasia in Inflammatory Bowel Diseases: Expert Review”, “Third European Evidence-based Consensus on Diagnosis and Management of Ulcerative Colitis. Part 1: Definitions, Diagnosis, Extra-intestinal Manifestations, Pregnancy, Cancer Surveillance, Surgery, and Ileo-anal Pouch Disorders”.
(Answer) Thank you for your kind comment. We annotated and added AGA and ECCO guidelines to the manuscript.
- The sample size is too small to derive reliable results. Other limits of the study are the monocentric nature, the lack of randomization to compare results and use of a single endoscopist.
(Answer) We thank the reviewer for their comments. We reinforced the limitation section as follows. This study had several limitations. First, the sample size was relatively small, even though a sample size calculation was performed. Second, this study was conducted at a single academic institution by a single endoscopist. Third, the study design was not a randomized controlled trial, although the tandem endoscopic design is considered to be a reliable method for evaluating different diagnostic modalities. Therefore, one should be careful when interpreting or extrapolating these data. Further prospective studies with a larger population are needed.
- Finally, I don’t understand why the colon segments have been divided into three equal parts in the evaluation of surveillance.
(Answer) Our study was performed using a segmental tandem colonoscopy technique. Tandem endoscopic design is considered a reliable method for evaluating different techniques/technologies for detecting lesions during endoscopy. Usually, when performing the colonoscopy, technique A was finished with the colonoscope and then reinsertion was performed for technique B. However, re-insertion of the colonoscope tended to be difficult due to air inflation and the long procedure time. Therefore, the colon was divided into three segments and tandem endoscopy was performed. The following studies and others applied the segmental tandem colonoscopy technique. 1. van Rijn, J.C.; Reitsma, J.B.; Stoker, J.; Bossuyt, P.M.; van Deventer, S.J.; Dekker, E. Polyp miss rate determined by tandem colonoscopy: a systematic review. Am J Gastroenterol 2006, 101, 343-350. 2. Zhao, S.; Song, Y.; Wang, S.; Wang, R.; Feng, Z.; Gong, A.; Yang, X.; Pan, P.; Yao, D.; Zhang, J.; et al. Reduced Adenoma Miss Rate With 9-Minute vs 6-Minute Withdrawal Times for Screening Colonoscopy: A Multicenter Randomized Tandem Trial. Am J Gastroenterol 2022, doi:10.14309/ajg.0000000000002055. 3. Gilani, N.; Stipho, S.; Panetta, J.D.; Petre, S.; Young, M.A.; Ramirez, F.C. Polyp detection rates using magnification with narrow band imaging and white light. World J Gastrointest Endosc 2015, 7, 555-562. 4.Hong, S.N.; Choe, W.H.; Lee, J.H.; Kim, S.I.; Kim, J.H.; Lee, T.Y.; Kim, J.H.; Lee, S.Y.; Cheon, Y.K.; Sung, I.K.; et al. Prospective, randomized, back-to-back trial evaluating the usefulness of i-SCAN in screening colonoscopy. Gastrointest Endosc 2012, 75, 1011-1021.e1012. 5. Hong, S.N.; Sung, I.K.; Kim, J.H.; Choe, W.H.; Kim, B.K.; Ko, S.Y.; Lee, J.H.; Seol, D.C.; Ahn, S.Y.; Lee, S.Y.; et al. The Effect of the Bowel Preparation Status on the Risk of Missing Polyp and Adenoma during Screening Colonoscopy: A Tandem Colonoscopic Study. Clin Endosc 2012, 45, 404-411. 6. Avalos, D.J.; Jia, Y.; Zuckerman, M.J.; Michael, M.; Gonzalez-Martinez, J.; Mendoza-Ladd, A.; Garcia, C.J.; Sunny, J.; Delgado, V.C.; Hernandez, B.; et al. Segmental Withdrawal During Screening Colonoscopy Does Not Increase Adenoma Detection Rate. South Med J 2020, 113, 438-446, doi:10.14423/smj.0000000000001147.
When the colon was divided into three parts, the cecum and ascending colon were considered the proximal colon, the transverse and descending colon were considered the mid-colon, and the sigmoid colon and rectum were considered the distal colon. In healthy controls, the proximal and mid-colon was can be discriminated based on the hepatic flexure, which can be identified by the curve of the colon and the bluish hue of the liver. The boundary of the mid and distal colon can be divided by the sigmoid-descending (SD) junction, which is usually a colonic length of 30 cm from the anal verge. However, long-standing extensive UC induces shortening of the colon, disturbing clear discrimination of the colonic segments, especially the mid and distal colon. The mean length of the colon (mean endoscope introduction length) in the enrolled patients was 76.2 ± 10.2 cm. In addition, the lead pipe appearance of the colon makes it difficult to identify the bending of the SD junction. Therefore, we measured the colonic length and divided the colon into three equal lengths.
Reviewer 2 Report
A pertinent study, although numbers are limited.
The following is suggested to enhance the manuscript:
1. The limitations section should be improved
2. The discussion has scope for improvement. The following maybe added and discussed:
a. Buchner AM. The Role of Chromoendoscopy in Evaluating Colorectal Dysplasia. Gastroenterol Hepatol (N Y). 2017;13(6):336-347.
b. Gabbani T, Manetti N, Bonanomi AG, Annese AL, Annese V. New endoscopic imaging techniques in surveillance of inflammatory bowel disease. World J Gastrointest Endosc. 2015 Mar 16;7(3):230-6. doi: 10.4253/wjge.v7.i3.230. PMID: 25789093; PMCID: PMC4360441.
c. Buchner AM, Lichtenstein GR. Evaluation and Detection of Dysplasia in IBD: the Role of Chromoendoscopy and Enhanced Imaging Techniques. Curr Treat Options Gastroenterol. 2016 Mar;14(1):73-82. doi: 10.1007/s11938-016-0078-y. PMID: 26831292.
d. Tontini GE, Vecchi M, Neurath MF, Neumann H. Review article: newer optical and digital chromoendoscopy techniques vs. dye-based chromoendoscopy for diagnosis and surveillance in inflammatory bowel disease. Aliment Pharmacol Ther. 2013 Nov;38(10):1198-208. doi: 10.1111/apt.12508. Epub 2013 Oct 3. PMID: 24117471.
e. Imperatore N, Castiglione F, Testa A, De Palma GD, Caporaso N, Cassese G, Rispo A. Augmented Endoscopy for Surveillance of Colonic Inflammatory Bowel Disease: Systematic Review With Network Meta-analysis. J Crohns Colitis. 2019 May 27;13(6):714-724. doi: 10.1093/ecco-jcc/jjy218. PMID: 30597029.
f. Štefănescu D, Pereira SP, Filip MM, Săftoiu A, Cazacu S. Advanced Endoscopic Imaging Techniques for the Study of Colonic Mucosa in Patients with Inflammatory Bowel Disease. Rom J Intern Med. 2016 Jan-Mar;54(1):11-23. doi: 10.1515/rjim-2015-0050. PMID: 27141566.
g. Huguet JM, Ferrer-Barceló L, Suárez P, et al. Colorectal cancer screening and surveillance in patients with inflammatory bowel disease in 2021. World J Gastroenterol. 2022;28(5):502-516. doi:10.3748/wjg.v28.i5.502
3. A way forward section should also augment the article citing upcoming data
Author Response
Revision Point-by-Point
Thank you very much for your generous and discerning comments and kind suggestions regarding our manuscript. We made our best efforts to revise the manuscript according to your comments and recommendations. The point-by-point responses to your comments and suggestions are shown below.
Answers to comments from the Reviewer #2
1. The limitations section should be improved
(Answer) Thank you for your careful comments. We revised and reinforced the limitation section in the Discussion as follows:
2. The discussion has scope for improvement. The following maybe added and discussed:
- Buchner AM. The Role of Chromoendoscopy in Evaluating Colorectal Dysplasia. Gastroenterol Hepatol (N Y). 2017;13(6):336-347.
- Gabbani T, Manetti N, Bonanomi AG, Annese AL, Annese V. New endoscopic imaging techniques in surveillance of inflammatory bowel disease. World J Gastrointest Endosc. 2015 Mar 16;7(3):230-6. doi: 10.4253/wjge.v7.i3.230. PMID: 25789093; PMCID: PMC4360441.
- Buchner AM, Lichtenstein GR. Evaluation and Detection of Dysplasia in IBD: the Role of Chromoendoscopy and Enhanced Imaging Techniques. Curr Treat Options Gastroenterol. 2016 Mar;14(1):73-82. doi: 10.1007/s11938-016-0078-y. PMID: 26831292.
- Tontini GE, Vecchi M, Neurath MF, Neumann H. Review article: newer optical and digital chromoendoscopy techniques vs. dye-based chromoendoscopy for diagnosis and surveillance in inflammatory bowel disease. Aliment Pharmacol Ther. 2013 Nov;38(10):1198-208. doi: 10.1111/apt.12508. Epub 2013 Oct 3. PMID: 24117471.
- Imperatore N, Castiglione F, Testa A, De Palma GD, Caporaso N, Cassese G, Rispo A. Augmented Endoscopy for Surveillance of Colonic Inflammatory Bowel Disease: Systematic Review With Network Meta-analysis. J Crohns Colitis. 2019 May 27;13(6):714-724. doi: 10.1093/ecco-jcc/jjy218. PMID: 30597029.
- Štefănescu D, Pereira SP, Filip MM, Săftoiu A, Cazacu S. Advanced Endoscopic Imaging Techniques for the Study of Colonic Mucosa in Patients with Inflammatory Bowel Disease. Rom J Intern Med. 2016 Jan-Mar;54(1):11-23. doi: 10.1515/rjim-2015-0050. PMID: 27141566.
- Huguet JM, Ferrer-Barceló L, Suárez P, et al. Colorectal cancer screening and surveillance in patients with inflammatory bowel disease in 2021. World J Gastroenterol. 2022;28(5):502-516. doi:10.3748/wjg.v28.i5.502
3. A way forward section should also augment the article citing upcoming data
(Answer) Thank you for your insightful comments. We would like to answer specific comments #2 and #3 together. We wrote a new Discussion section citing the references you recommended and recently published data as follows: Previous studies demonstrated that DCE improved colorectal neoplasia detection in high-risk patients with long-standing IBD, and VCE has been shown to improve the characterization of diminutive colorectal lesions [1-6]. A recent network meta-analysis identified the significant superiority of DCE to WLE in detecting dysplasia (OR = 2.12), which was similar to our findings [3]. Because of their narrow scanning area, DCE and VCE are best used in conjunction with WLE in routine practice. The best combination for IBD surveillance appears to be WLE for the identification of suspicious areas, with a further examination by DCE to detect superficial colorectal neoplasia [4]. However, cost, availability, time, and experience are still issues. Therefore, evolving technology in combination with HD-WLE can be applicable in selected cases in routine clinical practice, enabling further definition of the lesions and the assessment of their histology, and thus, facilitates real-time in vivo diagnoses and decisions regarding the resection of lesions [7]. Furthermore, VCE techniques are superior to WLE for assessing the activity and extent of colorectal IBD, which can bring additional benefits in assessing mucosal healing [2].
Recently, novel endoscopy techniques have emerged [6]. Confocal laser endomicroscopy (CLE) can be used to characterize a lesion, providing the same results as conventional histology [5], so in vivo microscopic assessment of the colonic mucosa may be possible [7]. Devices to improve endoscopic stabilization and visualization, such as Endocuff, may facilitate the detection and removal of colorectal neoplasia, especially in the flexible folds of the sigmoid colon [8]. Recently, non-blinded and randomized controlled trials showed that computer-aided detection improved the detection of colorectal neoplasia by providing visual alarms during the procedure [9,10]. Most studies evaluating the efficacy of newly developed devices in improving the detection of colorectal neoplasia tended to exclude patients with IBD. Therefore, whether new technology, such as the Endocuff and computer-aided detection, improves UC surveillance should be studied.
References
- Buchner, A.M. The Role of Chromoendoscopy in Evaluating Colorectal Dysplasia. Gastroenterol Hepatol (N Y) 2017, 13, 336-347.
- Tontini, G.E.; Vecchi, M.; Neurath, M.F.; Neumann, H. Review article: newer optical and digital chromoendoscopy techniques vs. dye-based chromoendoscopy for diagnosis and surveillance in inflammatory bowel disease. Aliment Pharmacol Ther 2013, 38, 1198-1208, doi:10.1111/apt.12508.
- Imperatore, N.; Castiglione, F.; Testa, A.; De Palma, G.D.; Caporaso, N.; Cassese, G.; Rispo, A. Augmented Endoscopy for Surveillance of Colonic Inflammatory Bowel Disease: Systematic Review With Network Meta-analysis. J Crohns Colitis 2019, 13, 714-724, doi:10.1093/ecco-jcc/jjy218.
- Gabbani, T.; Manetti, N.; Bonanomi, A.G.; Annese, A.L.; Annese, V. New endoscopic imaging techniques in surveillance of inflammatory bowel disease. World J Gastrointest Endosc 2015, 7, 230-236, doi:10.4253/wjge.v7.i3.230.
- Štefănescu, D.; Pereira, S.P.; Filip, M.M.; Săftoiu, A.; Cazacu, S. Advanced Endoscopic Imaging Techniques for the Study of Colonic Mucosa in Patients with Inflammatory Bowel Disease. Rom J Intern Med 2016, 54, 11-23, doi:10.1515/rjim-2015-0050.
- Huguet, J.M.; Ferrer-Barceló, L.; Suárez, P.; Sanchez, E.; Prieto, J.D.; Garcia, V.; Sempere, J. Colorectal cancer screening and surveillance in patients with inflammatory bowel disease in 2021. World J Gastroenterol 2022, 28, 502-516, doi:10.3748/wjg.v28.i5.502.
- Buchner, A.M.; Lichtenstein, G.R. Evaluation and Detection of Dysplasia in IBD: the Role of Chromoendoscopy and Enhanced Imaging Techniques. Curr Treat Options Gastroenterol 2016, 14, 73-82, doi:10.1007/s11938-016-0078-y.
- van Doorn, S.C.; van der Vlugt, M.; Depla, A.; Wientjes, C.A.; Mallant-Hent, R.C.; Siersema, P.D.; Tytgat, K.; Tuynman, H.; Kuiken, S.D.; Houben, G.; et al. Adenoma detection with Endocuff colonoscopy versus conventional colonoscopy: a multicentre randomised controlled trial. Gut 2017, 66, 438-445, doi:10.1136/gutjnl-2015-310097.
- Wang, P.; Liu, P.; Glissen Brown, J.R.; Berzin, T.M.; Zhou, G.; Lei, S.; Liu, X.; Li, L.; Xiao, X. Lower Adenoma Miss Rate of Computer-Aided Detection-Assisted Colonoscopy vs Routine White-Light Colonoscopy in a Prospective Tandem Study. Gastroenterology 2020, 159, 1252-1261.e1255, doi:10.1053/j.gastro.2020.06.023.
- Wan, J.; Zhang, Q.; Liang, S.H.; Zhong, J.; Li, J.N.; Ran, Z.H.; Zhi, F.C.; Wang, X.D.; Zhang, X.L.; Wen, Z.H.; et al. Chromoendoscopy with targeted biopsies is superior to white-light endoscopy for the long-term follow-up detection of dysplasia in ulcerative colitis patients: a multicenter randomized-controlled trial. Gastroenterol Rep (Oxf) 2021, 9, 14-21, doi:10.1093/gastro/goaa028.
Reviewer 3 Report
General comments: Patients with ulcerative colitis (UC) present with a higher risk of colorectal cancer(CRC) development. Current guidelines recommend dysplasia surveillance
with dye-spraying chromoendoscopy (DCE) and/or narrow-band imaging (NBI)
techniques. Biopsies directed to abnormal mucosa with an aid of DCE or NBI are
considered the preferred surveillance methods in comparison with random biopsies.
Nevertheless, the rate of dysplasia and sessile serrated adenoma detection by using
the aforementioned techniques is unclear. Therefore, the subject of the present
manuscript is interesting, however, some issues have to be addressed.
Specific comments:
The additional post-review file has been attached with some tips.
Parts that require English language correction were highlighted in yellow and parts that need substantive corrections are in green.
My additional comments are as follows:
Line 71 „≥ 8 years since symptom occurrence”- change to: since UC diagnosis
„clinical remission (partial Mayo score ≤ 2)- partial Mayo score 2 does not always present with clinical remission. As the Authors presented in Table 1, there were no patients with partial Mayo score 2 in the study group.
Line 74 „6) coagulopathy”- how was coagulopathy defined?
Line 80 „the colonic length was measured.”- what was the purpose of measuring the colonic length? Cecum intubation should be confirmed by visualization of the appendix and ileocecal valve, not by the colonic length.
Line 153 „The mean colonic length was 76.2 ± 10.3 cm.” Rather mean endoscope introduction length.
Line 154 „The withdrawal time of HD-WLE, HD-NBI, and HD=DCE was 6.5 ± 1.1, 6.2 ± 2.0, 154 and 9.2 ± 3.4 minutes, respectively” I understand that the withdrawal time excluding the time of biopsies and polypectomies.
Line 162 „In colitis involving the colon…” Colitis always affects the colon
English polishing is needed.
My conclusion: revision is required before the final decision of paper acceptance for publication.

Author Response
Revision Point-by-Point
Thank you very much for your generous and discerning comments and kind suggestions regarding our manuscript. We made our best efforts to revise the manuscript according to your comments and recommendations. The point-by-point responses to your comments and suggestions are shown below.
Answers to comments from the Reviewer #3
- The additional post-review file has been attached with some tips. Parts that require English language correction were highlighted in yellow and parts that need substantive corrections are in green.
(Answer) We thank the reviewer for pointing this out, and we agree with the reviewer. Therefore, we asked for help from a professional English editing service (Harrisco). The certificate of proofreading is attached.
- Line 71 „≥ 8 years since symptom occurrence”- change to: since UC diagnosis
(Answer) Thank you for your kind comment. We agree with the reviewer’s comments. All included patients were ≥ 8 years from symptom occurrence and UC diagnosis. We revised the text as suggested by the reviewer.
- „clinical remission (partial Mayo score ≤ 2)- partial Mayo score 2 does not always present with clinical remission. As the Authors presented in Table 1, there were no patients with partial Mayo score 2 in the study group.
(Answer) Thank you for your kind comment. We agree with the reviewer’s comments that a partial Mayo score of 2 does not always present with clinical remission. We changed the definition of clinical remission from a partial Mayo score of ≤ 2 to a partial Mayo score of < 2, as pointed out by the reviewer.
- Line 74 „6) coagulopathy”- how was coagulopathy defined?
(Answer) Before endoscopy, all patients underwent history-taking, complete blood count (CBC), and prothrombin time (PT) with international normalized ratio (INR). We defined coagulopathy as patients taking dual anti-platelet agents or warfarin, a platelet count of < 100,000/mm3, and/or PT INR > 1.4. We added this point to section 2.1 Study Population as follows: 6) coagulopathy (patients taking dual anti-platelet agents and warfarin, platelet count of < 100,000/mm3, and/or prothrombin time international normalized ratio > 1.4).
- Line 80 „the colonic length was measured.”- what was the purpose of measuring the colonic length? Cecum intubation should be confirmed by visualization of the appendix and ileocecal valve, not by the colonic length.
(Answer) We thank the reviewer for pointing this out. We agree with the reviewer’s comments that cecum intubation must be confirmed by visualization of the appendix and ileocecal valve, not colonic length.
When the colon was divided into three parts, the cecum and ascending colon were considered the proximal colon, the transverse and descending colon were considered the mid-colon, and the sigmoid colon and rectum were considered the distal colon. In healthy controls, the proximal and mid-colon can be discriminated based on the hepatic flexure, which can be identified by the curve of the colon and the bluish hue of the liver. The boundary of the mid and distal colon can be divided by the sigmoid-descending (SD) junction, which usually is a colonic length of 30 cm from the anal verge. However, long-standing extensive UC induces shortening of the colon length, disturbing clear discrimination of the colonic segments, especially the mid and distal colon. The mean length of the colon (mean endoscope introduction length) in the enrolled patients was 76.2 ± 10.2 cm. In addition, the lead pipe appearance of the colon makes it difficult to identify the bending of the SD junction. Therefore, we measured the colonic length and divided the colon into three equal lengths.
We described this in the Methods section as follows: The colonoscope was advanced to the cecal end and colonic length was measured. However, the phrase “the colonic length was measured” can be misinterpreted, and we removed it.
- Line 153 „The mean colonic length was 76.2 ± 10.3 cm.” Rather mean endoscope introduction length.
(Answer) We thank the reviewer for pointing this out. We agree with the reviewer’s comment. We estimated the colonic length when the colonoscope was intubated into the cecum. During the colonoscopy, the colon was distended and looping. Therefore, as the reviewer mentioned, the description “mean endoscope introduction length” was more reasonable. However, there was a possibility of misinterpretation. So, we removed “the mean colonic length was 76.2 ± 10.3 cm” from the table
- Line 154 „The withdrawal time of HD-WLE, HD-NBI, and HD-DCE was 6.5 ± 1.1, 6.2 ± 2.0, and 9.2 ± 3.4 minutes, respectively” I understand that the withdrawal time excluding the time of biopsies and polypectomies.
(Answer) I understand that the reviewer is concerned with the withdrawal time of each technique. As the reviewer commented, we measured the withdrawal time from the colon using a stopwatch, which excluded the time used for washing, biopsy, and polypectomy. Therefore, the pure observation time for HD-WLE and HD-NBI during colonoscopy withdrawal was just over six minutes. However, in practice, withdrawal time includes the time for washing, biopsy, and polypectomy during colonoscopy withdrawal, and this total withdrawal time was 33.5 ± 12.1 minutes. When simply calculated, the withdrawal time for HD-WLE, HD-NBI, and HD-DCE can be estimated to be more than 10 minutes. We described this point in the limitations of the study in the Discussion section.
- Line 162 „In colitis involving the colon…” Colitis always affects the colon
(Answer) We thank the reviewer for their thoughtful comments. We agree with the reviewer’s comment. We just wanted to emphasize the exclusion of sporadic lesions that developed in the colonic area without UC involvement. We changed “In colitis involving the colon” to “In the UC-involved colonic area.”
- English polishing is needed.
(Answer) We thank the reviewer for pointing this out, and we agree with the reviewer. Therefore, we asked for help from a professional English editing service (Harrisco). The certificate of proofreading is attached.